# Management of Patients with Lower-Risk Myelodysplastic Neoplasms (MDS)

Josephine Lucero [1] , Salman Al-Harbi [1] and Karen W. L. Yee [1,2,*]

1    Division of Medical Oncology and Hematology, Princess Margaret Cancer Centre,
     University Health Network, 700 University Avenue, 6th Floor, Toronto, ON M5G 1Z5, Canada;
     josephine.lucero@uhn.ca (J.L.); salman.alharbi@uhn.ca (S.A.-H.)
2    Division of Hematology, University of Toronto, Toronto, ON M5S 3H2, Canada
*    Correspondence: karen.yee@uhn.ca; Tel.: +1-416-946-4495; Fax: +1-416-946-4563

**Abstract:** Myelodysplastic neoplasms (MDS) are a heterogenous group of clonal hematologic disorders characterized by morphologic dysplasia, ineffective hematopoiesis, and cytopenia. In the past year, the classification of MDS has been updated in the 5th edition of the World Health Organization (WHO) Classification of Haematolymphoid Tumours and the International Consensus Classification (ICC) of Myeloid Neoplasms and Acute Leukemia with incorporation of morphologic, clinical, and genomic data. Furthermore, the more comprehensive International Prognostic Scoring System-Molecular (IPSS-M) allows for improved risk stratification and prognostication. These three developments allow for more tailored therapeutic decision-making in view of the expanding treatment options in MDS. For patients with lower risk MDS, treatment is aimed at improving cytopenias, usually anemia. The recent approval of luspatercept and decitabine/cedazuridine have added on to the current armamentarium of erythropoietic stimulating agents and lenalidomide (for MDS with isolated deletion 5q). Several newer agents are being evaluated in phase 3 clinical trials for this group of patients, such as imetelstat and oral azacitidine. This review provides a summary of the classification systems, the prognostic scores and clinical management of patients with lower risk MDS.

**Keywords:** myelodysplastic neoplasm; prognostic scoring system; lenalidomide; luspatercept; antithymocyte globulin; hypomethylating agent; imetelstat

## 1. Introduction

Myelodysplastic neoplasms (MDS) are a heterogenous group of clonal hematologic disorders characterized by morphologic dysplasia, ineffective hematopoiesis, and cytopenias. With recent updates to classification and prognostic systems, alongside improvements in treatment, therapeutic strategies will become more personalized in the coming years. We review the recent changes to the classification systems, prognostic scores, and discuss the expanding therapeutic armamentarium for treating adult patients with lower-risk (LR) MDS as defined by IPSS low and intermediate-1-risk and revised IPSS (IPSS-R) very low, low and intermediate-risk ($\leq$3.5) categories.

## 2. Classifications

Two updated classifications of myeloid malignancies were introduced in 2022: (a) the 5th edition of the World Health Organization classification (WHO 2022) and (b) the International Consensus Classification (ICC 2022). Both continue to expand the disease types that are genetically defined (Table 1) [1–3]. The classifications attempt to further refine and subtype this heterogeneous disease. WHO 2022 introduced the term myelodysplastic neoplasm (abbreviated MDS) to replace myelodysplastic syndrome, to emphasize the neoplastic nature of the disease [1,2]. MDS entities are grouped into those having defining genetic abnormalities and those that are morphologically defined, allowing utilization of

more comprehensive risk stratification schemes, such as the Revised International Prognostic Scoring System (IPSS-R), for improved prognostication. A consistent definition of cytopenia is used in both MDS and clonal cytopenia of undetermined significance (CCUS) and the threshold for dysplasia is maintained at 10%. Unlike in the WHO 2016 classification, WHO 2022 does not recognize certain cytogenetic abnormalities being MDS-defining, in the absence of morphologic dysplasia [4–6]. The distinction between single lineage (SLD) and multi-lineage dysplasia (MLD) is now considered optional. Hypoplastic MDS (MDS-h) is recognized as a new distinct subtype. The term MDS-excess blasts (MDS-EB) has been replaced by MDS-increased blasts (MDS-IB) compared to MDS-low blasts (<5% blasts; MDS-LB) for better clarification, while maintaining the long-standing cutoff of 10% to distinguish MDS-IB1 and MDS-IB2. This differs from ICC 2022 where MDS-EB2 has been changed to MDS/acute myeloid leukemia (AML) with 10–19% blasts. WHO 2022 softens the boundaries between MDS and AML, but the 20% blast cut-off is retained. WHO 2022 is concerned that (a) lowering the blast cut-off is arbitrary and does not reflect the biologic continuity in myeloid pathogenesis; (b) blast enumeration is subjective and prone to sampling variations/error; (c) no gold standard exists for blast enumeration; and (d) there is a risk of overtreatment if the blast count is lowered. However, there is broad agreement that MDS-IB2 may be regarded as AML-equivalent for therapeutic consideration and clinical trial design.

ICC 2022 also maintains the threshold of dysplasia at 10%, with a higher threshold being warranted when dysmegakaryopoeisis, other than micromegakaryocytes, are included [3]. Similar to WHO 2022, the definition of cytopenia is consistent in MDS and CCUS. ICC 2022 recognizes the following MDS-defining cytogenetic abnormalities, irrespective of dysplasia, in the context of persistent cytopenia: del(5q), multi-hit *TP53* mutation, and −7/del(7q) and complex karyotype (>3 unrelated clonal chromosomal abnormalities in the absence of other class-defining recurring genetic abnormalities), which are classified as MDS with del(5q), MDS with mutated *TP53*, or MDS, not otherwise specified (MDS, NOS), respectively. In the absence of clonality, the diagnosis of MDS requires the presence of qualifying dysplasia and persistent cytopenia.

Both WHO 2022 and ICC 2022 recognize 3 subtypes with MDS-defining genetic abnormalities:

(a) MDS with del(5q), whose definition has not changed, but thrombocytosis (platelet > $450 \times 10^9$/L) is permitted.

(b) MDS-*SF3B1* is a distinct disease that includes >90% of MDS cases with ≥5% ring sideroblasts (RS). WHO 2022 includes cases with *SF3B1* wildtype and RS > 15% in this category to allow inclusion of driver mutations in other RNA splicing components. Patients with low blasts and ≥15% RS without *SF3B1* mutation account for 3–4% of all MDS cases. In contrast, ICC 2022 excludes cases without *SF3B1* mutation in this category as SF3B1-unmutated MDS-RS cases have clinical features and outcomes similar to MDS with SLD or MLD and are now classified as MDS, NOS, irrespective of the number of RS.

(c) MDS with biallelic (or multihit) *TP53* alterations (MDS-bi*TP53*) consists of cases with >2 mutations of *TP53* or a *TP53* mutation with concurrent *TP53* copy loss or copy neutral loss of heterozygosity (e.g., deletion of the other allele on chromosome 17p). *TP53* alterations are biallelic in about two-thirds of MDS cases with *TP53* alterations. Over 90% of MDS-bi*TP53* have complex cytogenetics and regarded as very high risk. Some data suggests that MDS-bi*TP53* may be regarded as an AML-equivalent [7,8].

**Table 1.** Changes in MDS (Myelodysplastic Neoplasms) Classification.

| WHO 2016 (4th ed) | WHO 2022 (5th ed) | ICC 2022 |
|---|---|---|
| MDS with single lineage dysplasia (MDS-SLD) | | MDS with mutated *SF3B1* |
| MDS with multilineage dysplasia (MDS-MLD) | | MDS with del(5q) |
| MDS with ring sideroblasts (MDS-RS)<br>    MDS-RS-SLD<br>    MDS-RS-MLD | **MDS with defining genetic abnormalities**<br>    MDS with low blasts & isolated 5q deletion (MDS-5q)<br>    MDS with low blasts & *SF3B1* mutation (MDS-*SF3B1*) [a]<br>    MDS with biallelic *TP53* inactivation (MDS-bi*TP53*) | MDS with mutated *TP53* |
| MDS with isolated del(5q) | | MDS, not otherwise specified (MDS, NOS)<br>    MDS, NOS without dysplasia<br>    MDS, NOS with single lineage dysplasia<br>    MDS, NOS with multilineage dysplasia |
| MDS with excess blasts (MDS-EB)<br>    MDS-EB-1<br>    MDS-EB-2 | **MDS, morphologically defined**<br>    MDS with low blasts (MDS-LB)<br>    MDS, hypoplastic (MDS-h)<br>    MDS with increased blasts (MDS-IB)<br>        MDS-IB1<br>        MDS-IB2<br>    MDS with fibrosis (MDS-f) | MDS with excess blasts |
| MDS, unclassifiable (MDS-U) | | MDS/AML [b]<br>    MDS/AML with mutated *TP53*<br>    MDS/AML with myelodysplasia-related gene mutations<br>    MDS/AML with myelodysplasia-related cytogenetic abnormalities<br>    MDS/AML, NOS |

ICC—International Consensus Classification; WHO—World Health Organization. [a] Detection of ≥15% ring sideroblasts may substitute for SF3B1 mutation (in cases with wildtype SF3B1 and >15% ring sideroblasts). Acceptable related terminology: MDS with low blasts and ring sideroblasts; [b] the previous category of MDS-EB2 with >10% blasts is changed to MDS/AML, defined as a cytopenic myeloid neoplasm and 10% to 19% blasts in the blood or BM.

## 3. Risk Assessment

MDS subtypes exhibit different rates of leukemia transformation and overall survival (OS). Prognostic scores are essential tools to predict risk of progression to AML and long-term outcomes (Table 2). Treatment decisions are largely guided by these prognostic risk scores. The IPSS was the first important standard used in determining prognosis for untreated patients with MDS [9]. In 2012, the IPSS-R demonstrated improved predictive power by refining marrow blast categories and depth of the cytopenias and allocated more weight to cytogenetic abnormalities [10]. However, both the IPSS and IPSS-R were developed using the French-American-British (FAB) classification, which utilized morphology and immunohistochemistry to define disease sub-types, and using data from treatment-naïve patients. Both scores do not include patients with therapy-related and secondary MDS or other genetic changes which affect outcome. These limitations may contribute, in part, to the large heterogeneity in outcomes observed within the IPSS-R intermediate-risk category [11,12]. Furthermore, both are not dynamic scoring systems.

**Table 2.** Comparison of MDS Prognostic Scoring Systems.

| | IPSS (Greenberg 1997) [9] | IPSS-R (Greenberg 2012) [10] | IPSS-M (Bernard 2022) [13] |
|---|---|---|---|
| Includes CMML | Yes [a] (if WBC $\leq 12 \times 10^9$/L) | Yes [b] (if WBC $\leq 12 \times 10^9$/L) | Yes [c] (if WBC $< 13 \times 10^9$/L) |
| Includes secondary/ therapy-related MDS | No | No | Yes [d] |
| Includes previously treated patients | No | No | Yes |
| Sensitivity to degree of cytopenias | Limited | Anemia, thrombocytopenia & neutropenia | Anemia & thrombocytopenia [e] |
| Range of karyotypes | 3 categories | 5 categories | 5 categories |
| Marrow blasts | <30% [a] | <30% [b] | <20% |
| Includes gene mutations | No | No | Yes (31) |
| Number of prognostic variables | 3 | 5 | 5 [f] |
| Number of risk groups | 4 | 5 | 4 |

[a] Including 15% CMML and 8% FAB RAEB-T (AML with 20–30% blasts by WHO classification); [b] Including 9% CMML and 6% FAB RAEB-T (AML with 20–30% blasts by WHO classification); [c] Including 9.5% CMML and 3% MDS/MPN-RS-T and MDS/MPN-U; [d] 8% were secondary/therapy-related; [e] ANC had a small weight in the IPSS-R model but was not independently prognostic in the IPSS-M model; [f] the 31 genes are counted as one variable.

The recently published Molecular IPSS (IPSS-M) was developed using the WHO classification and incorporated hematologic parameters, the IPSS-R cytogenetic risk groups, as well as 31 somatic gene mutations [13]. In the discovery cohort, 3% of patients had DDX41 mutations, of which 87% likely had a germline variant, and 30% received treatment (62% hypomethylating agents [HMAs]; 7% intensive chemotherapy; 20% lenalidomide; 30% hematopoietic stem cell transplantation [HCT]). Of the genetic mutations included, multihit TP53, FLT3-ITD and FLT3-TKD, and MLL partial tandem duplication (PTD) were the top predictors of adverse outcome. In contrast, SF3B1 mutations were associated with favorable outcomes, but this was modulated by patterns of co-mutations. The IPSS-M demonstrated improved prognostic accuracy for OS, leukemia free survival (LFS) and AML transformation. The IPSS-M re-stratified 46% of patients in the IPSS-R categories (74% were upstaged and 26% were downstaged). It is applicable for patients with secondary or therapy-related MDS. Additionally, calculation of the IPSS-M allows for missing values with IPSS-M scores generated under the best, average and worst scenarios. Hence, if a significant number of values are missing, there can be a wide range in assigned risk groups

and outcomes. Most studies have used the IPSS and IPSS-R to risk stratify MDS patients for inclusion. The IPSS-M is currently undergoing validation.

## 4. Myeloid Malignancies with Germline Predisposition

Myeloid neoplasms with germline predisposition were first recognized as an entity in the WHO 2016 classification [4] and is retained in both WHO 2022 and ICC 2022 (Table 3) [1,3]. Both classifications recognize that there are other germline variants that predispose individuals to hematologic malignancies (such as CHEK2, Nijmegen breakage syndrome, and CSF3R) that are not included in the current subcategories [1,3,14]. In individuals with genetic conditions associated with an increased risk of hematologic malignancies, myeloid neoplasms with identified germline predispositions can occur. The frequency of pathogenic/likely pathogenic (P/LP) germline variants in MDS patients diagnosed at age < 40 years is 15–20% [15,16]. In MDS patients of all ages treated with allogeneic HCT, P/LP germline variants were found in 7% [17].

**Table 3.** Myeloid neoplasms with germline predispositions.

| Syndrome Name | Gene | Inheritance | Age of Onset | Predisposition to Other Cancers |
|---|---|---|---|---|
| **Myeloid neoplasms with germline predisposition without a preexisting platelet disorder or organ dysfunction** | | | | |
| Germline predisposition due to *CEBPA* P/LP variants | *CEBPA* | AD | Wide range | Not yet described |
| Germline predisposition due to *DDX41* P/LP variants | *DDX41* | AD | Adult > childhood | Likely |
| Li-Fraumeni syndrome | *TP53* | AD | Wide age range | Yes |
| **Myeloid neoplasms with germline predisposition and preexisting platelet disorders** | | | | |
| Germline predisposition due to *RUNX1* P/LP variants | *RUNX1* | AD | Wide age range | Myeloid malignancies > T-ALL > B cell malignancies |
| Germline predisposition due to *ANKRD26* P/LP variants | *ANKRD26* | AD | Adult > childhood | Not yet described |
| Germline predisposition due to *ETV6* P/LP variants | *ETV6* | AD | Wide age range | ALL > myeloid malignancies |
| **Myeloid neoplasms with germline predisposition and potential organ dysfunction** | | | | |
| Germline predisposition due to *GATA2* P/LP variants | *GATA2* | AD | Adolescents & young adults | Yes |
| Bone marrow failure syndromes: | | | | |
| Severe congenital neutropenia | *ELANE, G6PC3GFI1, HAX1, JAGN, TCRG1, VPS45A* | AD, AR | Adolescents & young adults | Not yet described |
| Shwachman-Diamond syndrome | *SBDS* (>90%), *DNAJC21, EFL1, SRP54* | AR | Childhood > adult | Not yet described |
| Fanconi anemia | *FANC A-W* | AR | Childhood > adult | Yes |
| Telomere biology disorders/short telomere syndromes | *ACD, CTC1, DKC1, MDM4, RTEL1, TERC, TERT, TINF2, NHP2, NOP10, NPM1, PARN, WRAP53, RPA1, Apollo* | AD, AR, and X-linked | Wide age range | Yes |
| RASopathies: | | | | |
| CBL syndrome | *CBL* | AD | Early childhood | Not yet described |
| Noonan syndrome | *PTPN11, NRAS, KRAS* | AD | Early childhood | ALL, AML, various nonhematologic cancers |
| Neurofibromatosis type 1 | *NF1* | AD | Childhood > adult | Yes |
| Down syndrome | Trisomy 21 | - | - | AML > ALL |
| Germline predisposition due to *SAMD9* P/LP variants | *SAMD9* | AD | Childhood > adult | Not yet described |
| Germline predisposition due to *SAMD9L* P/LP variants | *SAMD9L* | AD | Childhood > adult | Not yet described |
| Bloom syndrome | *BLM* | AR | Childhood > adult | Yes |

Modified from Döhner et al. [14]. Copyright permission obtained from Elsevier. AD—autosomal dominant; ALL—acute lymphoblastic leukemia; AR—autosomal recessive; P/LP—pathogenic/likely pathogenic.

With the more widespread use of next generation sequencing (NGS), there is an increasing awareness that individuals may harbor a pathogenic germline mutation that predisposed them to the hematologic malignancy. Accurate identification of patients with germline predisposition disorder is important as it affects management (donor selection for allogeneic HCT, choice of conditioning regimen, and use of post-transplant cyclophosphamide), genetic counseling and surveillance for the affected individual and their family [18,19].

Universal germline predisposition testing in patients with myeloid neoplasms, regardless of age, is currently not standard of care. Guidelines focus on early identification of younger patients with myeloid neoplasms through personal and family history screening questions, as well as the identification of variants with germline potential on NGS gene panels [18–21]. There are limitations to testing only patients diagnosed with a myeloid neoplasm before the age of 40 or 50 years and/or with a strong personal or family history, as patients may be unaware of their family history and there can be a wide variation in the age of disease onset due to variable penetrance of the germline mutation (e.g., patho-genic DDX41 variants). Furthermore, although somatic NGS panels can detect gene variants that are suspicious for germline alterations, the genes being evaluated in institutional and/or commercial panels may not include all the genes or gene regions involved in germline predisposition disorders, copy number variants are not routinely tested, and there may be technical difficulties, such as with read depth [18,19].

## 5. Therapeutic Options

Treatment of LR MDS focuses on improving cytopenias to prevent complications and maintain quality of life (QoL). In patients with untreated LR MDS, death has been attributed to infections (38%), transformation to AML (15%) and bleeding (13%) [22]. Isolated neutropenia or thrombocytopenia is uncommon and is more commonly observed in more than one lineage [23–25]. Neutropenia is seen in nearly 50% of newly diagnosed patients with MDS, including 15–20% of LR MDS patients. In terms of the use of prophylactic granulocyte colony-stimulating factor (G-CSF), a Cochrane systematic review demonstrated a substantial lack of data for the prevention of infections, prolongation of survival and improvement in QoL [26]. However, G-CSF has been used transiently in patients with <5% marrow blasts who have significant infections.

The prevalence of thrombocytopenia in MDS patients ranges from 40% to 65% with severe thrombocytopenia ($<20 \times 10^9$/L) occurring in <20% of patients [25]. In patients with untreated LR MDS, 51% and 12% of patients had platelet counts $<100 \times 10^9$/L and $<20 \times 10^9$/L, respectively, compared with 77% and 29%, respectively, in those with HR MDS. Two thrombopoietin (TPO) receptor agonists (i.e., romiplostim and eltrombopag) have been evaluated in MDS patients. Romiplostim was evaluated in randomized, placebo-controlled, phase 2 study in patients with LR MDS and thrombocytopenia [27,28]. Although romiplostim did not reduce the incidence of clinically significant bleeding events ($p = 0.13$), protocol-defined platelet transfusions were significantly reduced ($p < 0.0001$) and platelet responses were also higher in the romiplostim arm (36.5% vs. 3.6%, $p < 0.001$). The study was terminated because of concerns that the transient increases in peripheral blasts observed with romiplostim put patients at risk for progression to AML. Five-year follow-up for transformation to AML and death did not differ between the 2 groups [29]. A subsequent trial of romiplostim in LR MDS with <5% marrow blasts demonstrated a hematologic improvement-platelet response (HI-P) rate of 42% with a median response duration of 48.5 weeks [30]. Predictors of response were SRSF2 mutation status and baseline hemoglobin levels, but not endogenous TPO levels or platelet transfusion history.

Eltrombopag has also been assessed in a randomized, placebo-controlled, single blinded study in patients with LR MDS and thrombocytopenia [28]. Higher platelet responses (42.3% vs. 11.1%, $p < 0.001$) and decreased bleeding events (19.8% vs. 31.5%, $p = 0.0002$) were observed in the eltrombopag arm compared to placebo. There was no difference in AML transformation between the 2 groups (9% vs. 7%, $p = 0.729$). Both agents

are approved in the United States (US), Canada and Europe for the treatment of chronic immune thrombocytopenia and in the case of eltrombopag, for severe aplastic anemia, but they are not approved for the treatment of thrombocytopenia in patients with LR MDS. However, both romiplostim and eltrombopag have been used off label to increase platelet counts and decrease bleeding events in patients with LR MDS with <5% blasts.

Anemia is the most common cytopenia in MDS patients, with >80% of patients being anemic at diagnosis [23,31,32]. Red blood cell (RBC) transfusions are the cornerstone of best supportive care (BSC). However, it is associated with iron overload which can affect organ function [33–35], transfusion reactions, and decrease in QoL, thus warranting the use of other therapeutic options (as discussed below). The medications for MDS patients have been approved based on studies defining LR MDS as IPSS low and intermediate-1 risk and/or IPSS-R very low, low and intermediate-risk (with some studies including IPSS-R scores of 4 to 4.5) (Table 4; Figure 1).

**Table 4.** Drug approvals for Lower-risk MDS.

| Drug | Indication | Regulatory Status | Reference |
|---|---|---|---|
| Azacitidine (AZA) | for the treatment of patients with the following [FAB] MDS subtypes: refractory anemia or refractory anemia with ring sideroblasts (if accompanied by neutropenia or thrombocytopenia and requiring transfusions), refractory anemia with excess blasts, refractory anemia with excess blasts in transformation, and chronic myelomonocytic leukemia (CMML) | FDA (2004) | Silverman 2002; Kornblith 2002; Silverman 2006 [36–38] |
| | for the treatment of adult patients with (a) IPSS Intermediate-2 and High-risk MDS and (b) AML with 20–30% blasts and multi-lineage dysplasia, according to the WHO classification [a,b] | FDA (expanded 2008); EMA (2008); HC (2009) | Fenaux 2009 [39] |
| Lenalidomide (LEN) | for the treatment of transfusion-dependent anemia in patients with IPSS Low or Intermediate-1 risk MDS with chromosome 5q deletion [c] | FDA (Sub-part H 2005); EMA (2013); HC (2008) | Fenaux 2011 [40] |
| Deferasirox (DFX) | for use in treating chronic iron overload due to transfusional hemosiderosis in patients ≥ 2 years of age | FDA (2005); EMA (2006) | Shashaty 2006; Cappellini 2006; Cappellini 2011 [41–43] |
| | for the management of chronic iron overload in patients with transfusion-dependent anemias aged ≥6 years and in patients aged 2 to 5 who cannot be adequately treated with deferoxamine | HC (2006) | |
| Decitabine (DEC) | for the treatment of adult patients with de novo or secondary MDS, untreated or previously treated with chemotherapy, including the following: (a) IPSS Intermediate-1, intermediate-2 and high-risk International Prognostic Scoring System (IPSS) groups and (b) all French-American-British (FAB) subtypes (refractory anemia, refractory anemia with ringed sideroblasts, refractory anemia with excess blasts, refractory anemia with excess blasts in transformation, and CMML) [a] | FDA (2006); HC (2019) | Kantarjian 2006 [44] |
| Decitabine/cedazuridine (DEC-C) | for the treatment of adult patients with de novo or secondary MDS, untreated or previously treated with chemotherapy, including the following: (a) IPSS Intermediate-1, intermediate-2 and high-risk International Prognostic Scoring System (IPSS) groups and (b) all French-American-British (FAB) subtypes (refractory anemia, refractory anemia with ringed sideroblasts, refractory anemia with excess blasts, and CMML) | FDA (2020); HC (2020) | Garcia-Manero 2019; Savona 2021 [45,46] |
| Luspatercept | for the treatment of anemia failing an ESA and requiring ≥2 RBC units over 8 weeks in adult patients with [IPSS-R] very low- to intermediate-risk MDS with ring sideroblasts (MDS-RS) [d] | FDA (2020); EMA (2020); HC (2021) | Fenaux 2020 [47] |

EMA—European Medicine Agencies; FDA—US Food and Drug Administration; HC—Health Canada. [a] HC approval only if patients are not considered candidates for HCT; [b] EMA approval only if patients are not considered candidates for HCT; [c] EMA approval only when other therapeutic options are insufficient or inadequate; [d] FDA approval also for patients with myelodysplastic/myeloproliferative neo-plasm with ring sideroblasts and thrombocytosis (MDS/MPN-RS-T).

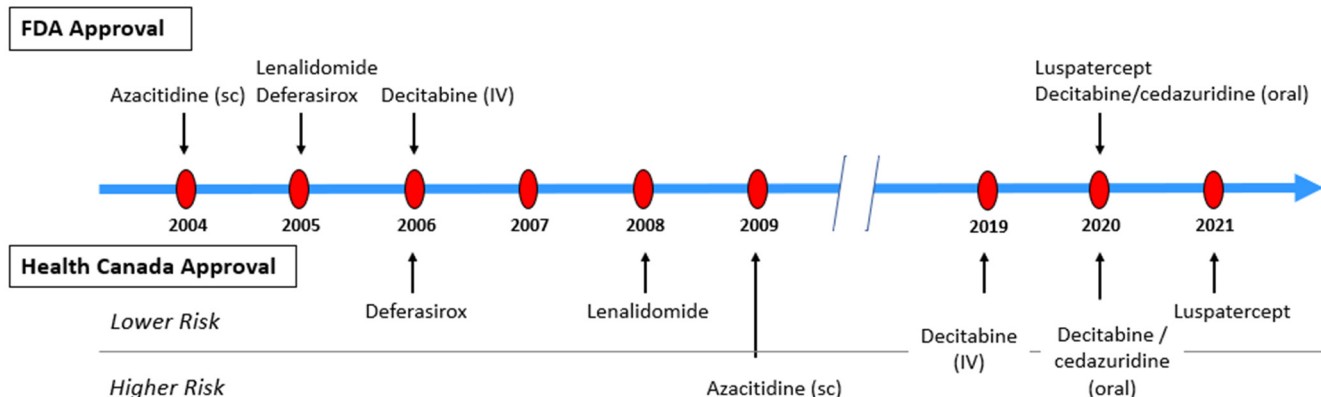

**Figure 1.** Drug Approval Timelines.

*5.1. Iron Chelation*

Deferasirox was approved for the treatment of chronic iron overload in patients with transfusion-dependent anemias based on data from clinical trials in patients with anemia due to a variety of disorders, including thalassemia and sickle cell disease [41–43]. The phase 2 TELESTO trial evaluated iron chelation using deferasirox in LR MDS patients with serum ferritin > 1000 ng/mL and transfusion history of 15 to 75 RBC units. It demonstrated a 36.4% risk reduction in event-free survival (EFS; defined by nonfatal events such as worsening cardiac function, hospitalization for congestive heart failure, liver impairment, cirrhosis, and transformation to AML or death). There was no effect on hemoglobin level, transfusion requirements or OS (although the median follow-up was only 2.4 years and the study was not powered to detect differences in OS) [48]. Study accrual was very slow, mainly as a result of deferasirox being approved and considered standard of care in some countries, and the study was changed from a phase 3 to a phase 2 study with the objective being altered from demonstrating superiority of iron chelation therapy (ICT) over placebo to evaluating clinical benefit (thus, allowing for a reduction in sample size).

Most guidelines, including those of the Canadian Consortium on MDS, recommend considering iron chelation for transfusion-dependent LR MDS patients with a serum ferritin > 1000 ng/mL, transfusion requirements approaching >20 units of RBCs and a life expectancy of >1–2 years or who are candidates for HCT [20,49].

*5.2. Erythropoiesis Stimulating Agents*

Erythropoiesis stimulating agents (ESA) decrease transfusion dependency and improve QoL, without increasing OS. Recombinant erythropoietin (EPO) and darbepoetin (DARBO) are the first-line agents used for treating anemia in LR MDS patients who have serum EPO (sEPO) levels $\leq$ 500 U/L. DARBO has a longer half-life due to its increased salicylate carbohydrate content. The overall response rate (ORR) to ESAs is 20–40% with most responses occurring within 3 months of treatment and response durations of 17 to 24 months [50]. Using the validated Nordic scoring system, patients with a sEPO level < 100 U/L and requiring <2 units of RBC per month, had a 74% probability of responding to ESAs [51].

Despite the widespread use of ESAs, phase 3 placebo-controlled trials have only recently been performed [52,53]. In the phase 3 study of EPO in ESA-naïve LR MDS patients with a low transfusion burden (TB), the response rate was higher in the EPO arm compared to placebo (31.8% vs. 4.4%, $p < 0.001$); all responders had sEPO < 200 U/L [52]. Although the HI-erythroid response (HI-E) was higher in the DARBO arm compared with placebo (14.7% vs. 0%, $p = 0.016$) [53], it was lower than that observed with EPO. The reason for the lower response rate with DARBO was attributed to administration every 3-weeks instead of every 2-weeks. Hence, European Medicines Agency (EMA) approved EPO, but not DARBO, for the treatment of anemia due to MDS in patients with a sEPO < 200 U/L. In the US and most provinces in Canada, approval of ESAs was for the management of chronic

anemia, and not specifically for anemia due to MDS. In patients who have never had or have lost a response to single agent ESA, the addition of G-CSF may rescue responses in up to 20% of cases, particularly in the presence of RS, without an increased risk of transformation to AML [54–56]. However, the dose/schedule of EPO administered in these studies differs from that used in current practice. For EPO, the study used an initial dose of 450 IU/kg (up to 40,000 IU total dose) with a provision to increase at week 8 to 1050 IU/kg (up to 80,000 IU total dose). In clinical practice, a starting weekly dose of a 40,000 IU is commonly used, increasing to a maximum of 80,000 IU depending on response [52]. For DARBO, the study used an every 3-weeks dosing, escalating to every 2 weeks at week 31. In clinical practice, dose escalation occurs in shorter intervals [53].

*5.3. Lenalidomide*

The improper regulation of the immune system is a significant factor in the development of MDS, leading to a failure in the production of healthy blood cells and contributing to the progression of the disease. [57,58]. The phase 2 MDS-003 study evaluated lenalidomide (LEN) in RBC transfusion-dependent LR MDS patients with del(5q) [59]. The response rate was 76% with 67% of patients achieving RBC transfusion independence (TI). Median time to response was 4.6 weeks and median duration of response was 2.2 years. Grade > 3 neutropenia and/or thrombocytopenia were manageable. The Food and Drug Administration (FDA) subsequently approved LEN for del(5q) LR MDS in 2005, followed by approval in Canada and Europe based on the confirmatory phase 3 study [40,60]. In the phase 3 study, patients were permitted to cross-over to LEN (5 mg or 10 mg) after 16 weeks. More patients who received LEN achieved RBC TI $\geq$ 26 weeks compared with placebo (42.6–56.1% vs. 5.9%, both $p < 0.001$). There was no difference in AML progression or OS; however, RBC TI $\geq$ 8 weeks was associated with prolonged OS and a trend toward reduced relative risk of AML progression [60]. Patients with del(5q) LR MDS who have received LEN and have evolution of pre-existing or emerging subclones with TP53 mutations are less sensitive to LEN and have a higher rate of disease progression, possibly due to the selective outgrowth of the TP53 mutated stem progenitor cells [61–64].

Preliminary results of a placebo-controlled, phase 3 study (Sintra-REV) evaluating LEN in non-transfusion-dependent, del(5q) LR MDS have been reported [65]. TP53 mutations were present in 17.2% and 27.8% of the LEN and placebo patients, respectively ($p = 0.48$). Early treatment with low dose (5 mg) LEN significantly prolonged the time to transfusion dependency (66.3 months vs. 11.6 months, $p = 0.021$) with improved HI-E (70% vs. 0%, $p < 0.001$) and cytogenetic responses (87.5% vs. 0%, $p < 0.001$) compared with placebo. After a 60-month follow-up, transfusion benefit was clear however, there was no difference in AML progression or OS.

LEN has also been evaluated in RBC transfusion-dependent, non-del(5q) LR MDS patients in the placebo-controlled, phase 3 MDS-005 study [66]. The response rate was 36.5% with RBC TI rate of 26.9% compared with 19.5% and 2.5%, respectively, in the placebo arm. Median duration of response was 30.9 weeks. The follow-up period was too short to permit comparison of AML progression between the 2 groups. Median OS was not reached. The most common side effects were neutropenia and thrombocytopenia. Regulatory approval has not been applied for this indication.

LEN can restore sensitivity to erythropoietin in MDS cells. Two open-label, phase 3 studies have assessed combination therapy with LEN and ESA compared to single agent LEN in transfusion-dependent, ESA-refractory, non-del(5q) LR MDS [67,68]. Both studies demonstrated higher response rates (28.3% vs. 11.5%, $p = 0.004$ and 39.4% vs. 23.1%, $p = 0.044$) with LEN and ESA combination therapy compared to LEN monotherapy. However, in both studies, there was a difference in response duration.

*5.4. Luspatercept*

Luspatercept is a recombinant fusion protein that binds transforming growth β superfamily ligands to reduce SMAD2 and SMAD3 signaling allowing better red cell maturation

by late-stage erythroblast differentiation [69]. The double-blind, placebo-controlled, phase 3 trial (MEDAL-IST) in patients with LR MDS with RS (including 17% with IPSS-R intermediate risk) demonstrated a higher rate of RBC TI for >8 weeks with luspatercept compared to placebo (38% vs. 13%, *p* < 0.001) [47]. Degree of TB affected duration and rates of RBC TI ≥ 8 weeks [70]. Median duration of RBC TI ≥ 8 weeks response was longer with luspatercept compared to placebo (29.9 weeks vs. 17.4 weeks). During weeks 1 to 48, 20.3% patients receiving luspatercept achieved >1 period of RBC TI ≥ 8 weeks response, including 33.3% LTB + ITD and 3% HTB patients. The trial was not powered to assess OS. Luspatercept was subsequently approved in the US for the treatment of transfusion-dependent patients with IPSS-R very low- to intermediate-risk MDS with RS or with MDS/MPN with RS and thrombocytosis who did not respond to, lost response to or are unlikely to respond to ESAs. In contrast, LEN approval is limited to MDS-RS patients in Canada and Europe.

As luspatercept can yield responses in LR MDS patients with <15% RS [71], a phase 3 study comparing luspatercept to ESA in ESA-naïve, transfusion-dependent, LR MDS patients (irrespective of RS status) is currently ongoing (COMMANDS, NCT03682536). The COMMANDS study met its primary endpoint (i.e., RBC TI for 12 weeks with a mean hemoglobin increase ≥1.5 g/dL), on a prespecified interim analysis, in 86 (59%) of 147 patients assigned to luspatercept versus 48 (31%) of 154 patients assigned to erythropoietin alfa therapy (common risk difference on response rate 26.6 [95% CI 15.8–37.4]; *p* < 0.0001) [72]. The median duration of RBC TI ≥ 12 weeks was longer with luspatercept than with epoetin alfa (127 weeks [95% CI 108–not estimable] vs. 77 weeks [39–not estimable]).

*5.5. Immunosuppressive Therapy*

Immunosuppressive therapy (IST) with antithymocyte globulin (ATG) or ATG with cyclosporine (CSA) yielded response rates of 24% to 49% [73–76]. A phase 3 trial comparing ATG + CSA with BSC demonstrated an ORR of 29% vs. 9% (*p* = 0.009) in favor of ATG + CSA [75]. Patients in the BSC arm were permitted to cross-over to the treatment arm. The study was limited by difficulties in accrual and inclusion of patients with IPSS higher risk disease and/or MDS with excess blasts. On analysis, only a hypocellular bone marrow (i.e., <20% cellularity) remained a significant predictor of achieving RBC TI [76].

*5.6. Hypomethylating Agents*
5.6.1. Azacitidine Injectable

Azacitidine (AZA) injectable was approved in the US for the treatment of patients with all FAB MDS subtypes (including refractory anemia with excess blasts in transformation and chronic myelomonocytic leukemia (CMML)). FDA approval was based on 2 single arm AZA studies [36] and a phase 3 Cancer and Leukemia Group B (CALGB) trial comparing AZA with BSC [37]. This phase 3 study demonstrated an ORR of 60% (7% complete response [CR]; 16% partial response [PR]; 37% HI) in the AZA arm compared to 5% (5% HI) in the BSC arm (*p* < 0.001) [37]. Treatment with AZA resulted in an improved QoL but did not demonstrate any improvement in OS (as patients in the BSC arm were permitted to cross-over and receive AZA after a minimum of 4 months) [37,38]. In Canada and Europe, AZA is only approved for IPSS HR patients with MDS as defined by the WHO classification [39].

5.6.2. Oral Azacitidine

Oral AZA has a unique pharmacologic profile from injectable AZA, and the 2 formulations are not bioequivalent [77,78]. In a randomized phase 3 trial, treatment with oral AZA significantly improved RBC TI rate compared with placebo (RBC TI for >56 days 30.8% vs. 11.1%, *p* = 0.0002; RBC TI for >84 days 28% vs. 5.6%, *p* < 0.0001) in RBC transfusion-dependent, HMA-naïve, LR MDS patients who were thrombocytopenic [79]. Median duration of 56-day RBC TI were 11.1 months and 5 months, respectively. The proportion of patients who had a >1.5 g/dL increase in hemoglobin concentration from baseline (23.4%

vs. 4.6%, *p* < 0.0001) and a HI-P (24.3% vs. 6.5%, *p* = 0.0003) were higher in the oral AZA arm. The overall death rate was similar, but there were more early deaths with oral AZA, driven by infections in those with pretreatment neutropenia. The study was terminated due to slow accrual but the primary endpoint was met: oral AZA was associated with a significantly higher rate of RBC TI compared with placebo. To mitigate the adverse events observed in previous trial, a phase 2/3 study comparing a different dose/schedule of oral AZA with placebo is currently being conducted in LR MDS patients who have at least one protocol-defined cytopenia (NCT05469737).

5.6.3. Decitabine Injectable

Decitabine (DEC) injectable was FDA approved in 2006 for the treatment of adult patients with treatment-naïve or previously treated, de novo or secondary, IPSS intermediate-1, intermediate-2 and high-risk MDS and CMML patients as defined by FAB classification, including patients with refractory anemia with excess blasts in transformation (i.e., AML with 20–30% blasts by WHO classification). An open-label, phase 3 study randomized patients with MDS as defined by FAB classification to either DEC 15 mg/m$^2$ intravenously [IV] every 8 h for 3 days every 6 weeks or BSC [44]. Treatment with DEC yielded a significantly higher ORR (17% vs. 0%, *p* < 0.001). Median duration of responses was 10.3 months. However, DEC did not prolong time to progression to AML or improve OS compared with BSC. The lower response rates and lack of survival benefit observed with DEC compared with AZA may be due to the DEC dose/schedule utilized in the study and the lower number of cycles of DEC administered compared with AZA (3 cycles vs. 9 cycles) [39]. Similar findings were observed in a phase 3 randomized trial in older patients with IPSS intermediate-1, intermediate-2 and high-risk MDS and CMML patients as defined by FAB classification [80]. ORR was 34% (13% CR, 6% PR and 15% HI) with DEC compared with 2% with BSC. Median number of cycles of DEC administered was 4. There was no improvement in OS or delay in time to progression to AML.

An alternative dose/schedule of DEC at 20 mg/m$^2$ IV daily for 5 days every 4 weeks yielded comparable response rates (ORR 32%; 17% CR, 15% marrow CR and 18% HI) with no new safety signals [81] and has been widely adopted. DEC is not approved in Europe for the treatment of patients with MDS, but it received Health Canada (HC) approval in 2019 for the same indications as in the US. However, DEC injectable is not being administered in Canada as the manufacturer has not applied for funding approval.

A randomized phase 2 study in patients with LR MDS and CMML compared lower doses of AZA (75 mg/m$^2$ IV/sc daily for 3 days) with lower doses of DEC (20 mg/m$^2$ IV daily for 3 days) every 28 days [82]. The ORR was 70% vs. 49% (*p* = 0.03) in favor of DEC compared with AZA. Thirty-two percent of patients treated with DEC became RBC TI compared with 16% with AZA (*p* = 0.2). As the higher ORR observed with DEC may be due to underdosing of AZA, a randomized phase 2 study comparing AZA for 3 or 5 consecutive days, DEC for 3 consecutive days every 28 days or BSC in patients with LR MDS and CMML was initiated (NCT02269280).

5.6.4. Oral Decitabine/Cedazuridine

Decitabine/cedazuridine (DEC-C) is an oral fixed-dose combination tablet of 35 mg decitabine and 100 mg cedazuridine, a cytidine deaminase inhibitor. Cedazuridine prevents the rapid metabolism of DEC in the gastrointestinal tract [83]. A phase 3 study in IPSS intermediate-1/2 and high risk MDS patients involved a crossover comparing 5 days of DEC-C with 5 days DEC IV every 28 days in the first 2 cycles. From cycle 3 onwards, all patients received DEC-C [45,46,84]. The study met its primary outcome: the 5-day cumulative area under the curve of DEC was similar in both arms. The ORR was 61.7% (22% CR; 32.3% marrow CR; 7.5% HI) with a median CR duration of 14 months. The safety profile was similar. After a median follow-up of 32 months, the median OS was 31.7 months. Both the FDA and HC have approved DEC-C for the treatment of adult patients with treatment-naïve or previously treated, de novo or secondary, IPSS intermediate-1, intermediate-2 and

high-risk MDS and CMML patients as defined by the WHO classification. DEC-C is not approved for patients with AML with 20–30% blasts.

A phase 1 study of DEC-C in LR MDS patients evaluated 6 different (lower) dose/schedules of DEC-C [85]. Adverse events were similar to those reported for standard dose DEC-C. HI was observed in 29.8% of patients. The recommended phase 2 dose (RP2D) of 10 mg DEC/100 mg cedazuridine daily for 5 days is being compared to 35 mg DEC/100 mg cedazuridine for 3 days in a 28-day cycle in an ongoing Phase 2 study (NCT03502668).

*5.7. Allogeneic Hematopoietic Stem Cell Transplant*

Allogeneic HCT is the only potentially curative therapy for patients with MDS but is associated with treatment-related morbidity and mortality. There are no prospective studies or good quality data evaluating the role of allogeneic HCT in patients with LR MDS [86,87]. Two decision analyses using a Markov decision model to determine timing of allogeneic HCT have been performed by the International Bone Marrow Transplant Registry (IBMTR) using retrospective data from patients with MDS stratified by the IPSS risk: (a) patients aged <60 years using myeloablative conditioning regimens (MAC) for HLA-matched sibling donor HCT with data obtained during the period 1990–1999 compared to a nontransplant cohort (treated predominantly with BSC) and (b) patients aged 60 to 70 years using reduced intensity conditioning regimens (RIC) for HLA-matched related or unrelated donor HCT with data obtained during the period 1998–2009 compared to a nontransplant cohort (treated with BSC, ESA or HMAs) [88,89]. Both indicated a benefit for upfront allogeneic HCT in patients with IPSS intermediate-2 and high risk MDS. However, for LR MDS, delayed transplantation but before transformation, was favored for individuals < 60 years and non-transplantation approaches for individuals aged 60 to 70 years.

The Gruppo Italiano Trapianto di Midollo Osseo (GITMO) also performed a decision analysis using a multistate Markov approach to determine the optimal timing of allogeneic HCT in patients with MDS stratified according to IPSS-R [90]. Patients aged of 19 to 81 years using either MAC (64%) or RIC (36%) for HLA-matched sibling or unrelated donor or single antigen mismatched unrelated donor (21%) HCT with data obtained during the period 1992–2010 compared to a nontransplant cohort (treated with BSC, ESA, HMAs and others). Life expectancy increased when transplantation was delayed from the initial stages to IPSS-R intermediate risk (gain-of-life expectancy 5.3, 4.7 and 2.8 years for patients aged ≤55, 60 and 65 years, respectively). Modeling decision analysis based on IPSS-R versus IPSS changed transplantation policy in a third of patients, resulting in a 2-year gain in life expectancy. All 3 decision models are hampered by the limited use of disease modifying agents such as HMAs, which are currently widely in use, and the lack of incorporation of molecular data.

At this time, transplantation should be considered, on a case-by-case basis, in appropriate patients with LR MDS who have poor risk features, such as persistent increased blasts, recurrent infections due to neutropenia, significant thrombocytopenia with bleeding and/or ongoing platelet transfusion requirements and/or lack of response to non-transplant therapies [91].

## 6. Newer Agents in Later Stages of Development

*6.1. Imetelstat*

Imetelstat is a 13-mer oligonucleotide targeting the RNA template of human telomerase. It is a first in-class competitive inhibitor of telomerase enzymatic activity. High telomerase activity has been seen in MDS and imetelstat targets cells with active telomerase. A phase 2/3 study (MDS3001) evaluating imetelstat in LR MDS patients who are relapsed/refractory or ineligible for ESAs is ongoing (NCT02598661). Data from the phase 2 part of this study have been reported [92]. Patients could not have received prior HMA or LEN. The 8- and 24-week RBC TI rate in the overall population was 37% and 23%, respectively, with median TI duration of 65 weeks. HI-E rate was 65% with a 63% reduction in RBC transfusion burden from baseline. A higher proportion of patients with

>50% reduction in expression of human telomerase reverse transcription (hTERT) achieved 8-week RBC TI. The most common adverse events were reversible cytopenias. Among responders, attainment of 24-week TI was predictive of a likelihood to achieve TI of more than 1 year [92,93].

Preliminary results of the randomized, placebo-controlled phase 3 portion of the trial were announced on 4 January 2023 (NCT02598661) [94]. The trial met its primary and secondary endpoints with improved 8-week RBC TI (39.8% vs. 15%, $p < 0.001$) and 24-week RBC TI (28% vs. 3.3%, $p < 0.001$) in patients receiving imetelstat compared to placebo, respectively. Median 8-week RBC TI duration approached 1 year for imetelstat compared to approximately 13 weeks for placebo ($p < 0.001$, HR = 0.23). Median 24-week RBC TI duration approached 1.5 years for imetelstat. RBC TI was achieved across all subtypes including those patients having RS and high or very high transfusion burden. The company is planning to submit for FDA approval in 2023.

### 6.2. Roxadustat

Roxadustat is an orally active and reversible inhibitor of hypoxia inducible factor prolyl hydroxylase (HIF-PH) [95,96]. Roxadustat thus prevents hydroxylation of HIF-$\alpha$ allowing for the transcription and expression of genes necessary for erythropoiesis. The open-label, dose selection, lead-in stage of the randomized phase 3, double-blind, placebo-controlled trial evaluating the efficacy and safety of roxadustat compared with placebo in red cell transfusion-dependent IPSS-R very low, low, and intermediate risk MDS has completed enrolment (NCT03263091) [97]. The lead-in stage was performed to determine the recommended phase 3 dose (RP3D) to be used for the phase 3 portion of the study. Forty-two percent of patients enrolled were ESA-naive and 58% of patients were ESA relapsed or refractory. RBC TI was achieved in 9 patients (37.5%) at 28 and 52 weeks. It was well tolerated with no fatalities or progression to AML. Roxadustat 2.5 mg/kg was chosen as the RP3D. The primary endpoint of the phase 3 portion of the study is RBC TI for >8 weeks. Enrolment has been completed; however, results have not been reported.

### 7. Treatment Algorithm for Lower-Risk MDS

All patients should receive supportive care based on symptoms, including RBC and platelet transfusions, G-CSF (if septic and/or recurrent severe infections), and antimicrobials as indicated (Figure 2). If the patient has progressed to a higher risk category, the patient should be transitioned to therapeutic options for HR MDS.

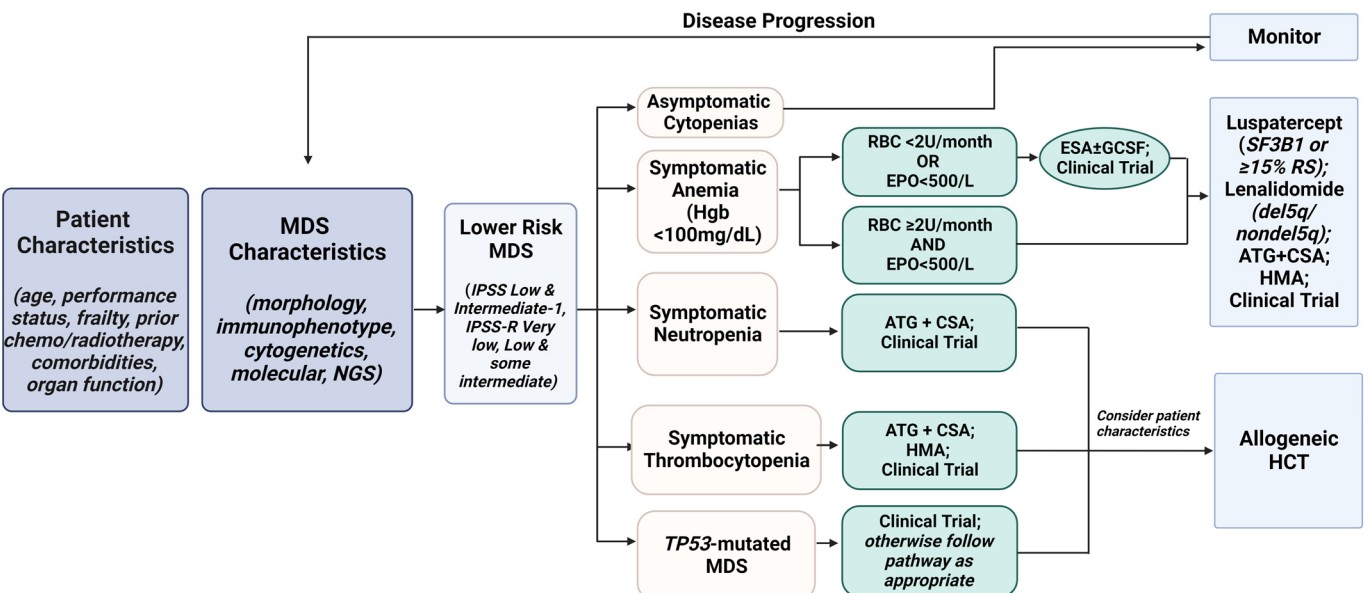

**Figure 2.** Lower Risk MDS: Treatment Algorithm.

If eligible and available, all patients should be considered for a clinical trial as there is an ongoing need to improve outcomes. Patients with biallelic TP53 mutations have a poor outcome with no effective therapies. The phase 3 trial of AZA with or without eprenetapopt in mono- and bi-allelic TP53 mutated MDS failed to meet its primary endpoint of CR rate with no difference in OS [98].

With increasing therapeutic options for patients with LR MDS, optimal sequencing of therapy needs to be addressed. Most of the studies that have led to drug approval have excluded patients who received prior therapy with LEN and HMAs. Retrospective studies suggest that the use of LEN before a HMA might be a better strategy than the reverse order [99,100]. Similarly, response to luspatercept after exposure to HMAs and LEN is lower than in patients who did not receive prior HMAs and/or LEN [101].

## 8. Conclusions

Both WHO 2022 and ICC 2022 emphasize genetically defined and morphologically defined MDS entities to facilitate diagnosis, prognostication and improve treatment. Although by definition, MDS-*SF3B1* and MDS-del(5q) (or MDS-5q) are lower risk entities and MDS-bi*TP53* (or MDS with mutated *TP53*) is a high-risk entity, these genetically defined subtypes make up a modest proportion of patients with MDS. Hence, both classifications yield risk stratification to more comprehensive risk stratification schemes, such as the IPSS-R, and possibly, in the future, to the IPSS-M. However, as more MDS entities become genetically defined, tailored therapy will improve patient outcomes and hopefully, minimize treatment-related toxicities.

The WHO 2016 category therapy-related myeloid neoplasm was removed from both WHO 2022 and ICC 2022 and replaced with diagnostic qualifiers to be added to the relevant MDS subtype. This permits the attribution of germline predisposition to development of MDS and cataloguing of MDS that arose following exposure to certain types of therapy (such as cytotoxic therapy or radiation therapy). This will enhance understanding of issues such as selection pressures of cytotoxic therapy on clonal hematopoiesis and risk of developing MDS. As more genetic information accumulates concerning MDS that arose after cytotoxic or radiation therapy, this will help answer the questions of whether a subset of these patients can be treated analogous to de novo MDS patients and will permit more tailored therapy for these patients. Similarly, with increasing awareness and screening for germline predisposition syndromes, it will be important to assess response to therapies in patients with MDS with germline predisposition.

It had been over a decade since a new drug was approved for the treatment of MDS, until the approval of luspatercept and DEC-C in 2020. It is likely that imetelstat in ESA failures and luspatercept in the upfront setting will be approved in the treatment of LR MDS. Other agents in later stages of development include oral AZA and roxadustat. Importantly, most of the studies that led to drug approval excluded patients who received prior LEN and/or HMA therapy and patients treated with imetelstat will not have received prior therapy with luspatercept. The increasing number of therapeutic options in LR MDS highlights the need to determine the optimal sequencing of therapies. It is an exciting and hopeful time for patients with LR MDS.

**Author Contributions:** J.L., S.A.-H. and K.W.L.Y. contributed to the writing, review and editing of the manuscript. All authors have read and agreed to the published version of the manuscript.

**Funding:** This research received no external funding.

**Conflicts of Interest:** K.W.L.Y. was a consultant for Bristol Myers Squibb/Celgene, F. Hoffmann-La Roche, GSK, Jazz Pharmaceuticals, Novartis, Pfizer, Shattuck Labs, Taiho Oncology, and Takeda Pharmaceutical Company; received research funding from Astex Pharmaceuticals, Forma Therapeutics, F. Hoff-mann-La Roche, Forma Therapeutics, Genentech, Geron Corporation, Gilead Sciences, Janssen Pharmaceuticals, Jazz Pharmaceuticals, Novar-tis, and Treadwell Therapeutics; and received honoraria from AbbVie and Novartis.

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
