# Peer review of "Management of Patients with Lower-Risk Myelodysplastic Neoplasms (MDS)"

_curroncol, doi:10.3390/curroncol30070459_

Round 1

Reviewer 1 Report

This paper reviewed the treatment methods for low-risk patients with myelodysplastic syndrome.

However, in the introduction, there is a description of an updated classification of myeloid malignancies by WHO and ICC , which feels disconnected from the paper topic. Although this update is a very big issue, I though it is not suitable as an introduction to the topic of this paper. It appears that the front part of the paper discusses the overall MDS, while the latter part focuses only on low risk. As a result, the content of the paper does not seem to be organically connected as a whole.

Furthermore, the definition of low risk presented in the paper is not clear. As the author mentioned, there are various risk prediction systems within the MDS. Therefore, it is necessary to specify which system classification the reviewed papers' low risk is based on.

Author Response

Please find below in bold the point-by-point replies to the comments:

  1. However, in the introduction, there is a description of an updated classification of myeloid malignancies by WHO and ICC , which feels disconnected from the paper topic. Although this update is a very big issue, I though it is not suitable as an introduction to the topic of this paper. It appears that the front part of the paper discusses the overall MDS, while the latter part focuses only on low risk. As a result, the content of the paper does not seem to be organically connected as a whole.

We feel including the 2 classifications is important to understand the significant changes in the diagnosis and treatment of MDS that have occurred in the last 1-2 years. We have tried to address this by including the comment in Section 8. Conclusion (lines 541-547):

Both WHO 2022 and ICC 2022 emphasize genetically defined and morphologically defined MDS entities to facilitate diagnosis, prognostication and improve treatment. Al -though by definition, the MDS-SF3B1 and MDS-del(5q) (or MDS-5q) are lower risk entities and MDS-biTP53 (or MDS with mutated TP53) is a high-risk entity, these genetically defined subtypes make up a modest proportion of patients with MDS. Hence, both classifications yield risk stratification to more comprehensive risk stratification schemes, such as the IPSS-R, and possibly, in the future, to the IPSS-M.

  1. Furthermore, the definition of low risk presented in the paper is not clear. As the author mentioned, there are various risk prediction systems within the MDS. Therefore, it is necessary to specify which system classification the reviewed papers' low risk is based on.

We had included the definition of lower risk MDS in Section 5. Therapeutic Options (lines 220-223):

The medications for MDS patients have been approved based on studies defining LR MDS as IPSS low and intermediate-1 risk and/or IPSS-R very low, low and intermediate-risk…

but have included it in Section 1. Introduction (lines 37-40) for clarity as suggested by the reviewer:

We review the recent changes to the classification systems, prognostic scores, and discuss the expanding therapeutic armamentarium for treating adult patients with lower-risk (LR) MDS as defined by IPSS low and intermediate-1-risk and revised IPSS (IPSS-R) very low, low and intermediate-risk (< 3.5) categories.”

Reviewer 2 Report

Myelodysplastic syndromes are a heterogeneous group of clonal disorders, in which peripheral cytopenia has an heavy impact on patients’ quality of life. In the past years many progress has been made on the pathogenesis of disease leading to the recent update of disease classification, and many new therapeutic options are at present available especially for high-risk MDS. In this paper the authors focused on  low risk disease, providing  an exhaustive overview of the current therapeutic options in this subset of MDS.

The paper is clear and updated. 

I have just few comments:

1.     Page 7 at the end of the section on erythropoiesis stimulating agents a more detailed comment on the discrepancy between EPO doses in clinical trials and in real life should be added;

2.     The recent WHO and ICC classifications introduced the MDS subtypes defined by genetic abnormalities. How could this change low-risk MDS management? Should de novo and therapy-related and secondary MDS differently managed? Pleas add a comment to conclusions

Author Response

Please find below in bold the point-by-point replies to the comments:

  1. Page 7 at the end of the section on erythropoiesis stimulating agents a more detailed comment on the discrepancy between EPO doses in clinical trials and in real life should be added;

The following has been added to Section 5.2 Erythropoiesis stimulating agents (lines 275-280):

For EPO, the study used an initial dose of 450 IU/kg (up to 40,000 IU total dose) with a provision to increase at week 8 to 1050 IU/kg (up to 80,000 IU total dose). In clinical practice, a starting weekly dose of a 40,000 IU is commonly used, increasing to a maximum of 80,000 IU depending on response [43]. For DARBO, the study used an every 3-weeks dosing, escalating to every 2 weeks at week 31. In clinical practice, dose escalation occurs in shorter intervals [44].

  1. The recent WHO and ICC classifications introduced the MDS subtypes defined by genetic abnormalities. How could this change low-risk MDS management? Should de novo and therapy-related and secondary MDS differently managed? Pleas add a comment to conclusions

There are discordant reports published about outcomes of patients with MDS arising from radiation therapy and/or cytotoxic therapy. This may be due in part to genetic changes observed and the currently available treatment options. This is beyond the scope of the current review. However, the following has been added to/modified in Section 8. Conclusion (lines 541-561):

Both WHO 2022 and ICC 2022 emphasize genetically defined and morphologically defined MDS entities to facilitate diagnosis, prognostication and improve treatment. Al -though by definition, the MDS-SF3B1 and MDS-del(5q) (or MDS-5q) are lower risk entities and MDS-biTP53 (or MDS with mutated TP53) is a high-risk entity, these genetically defined subtypes make up a modest proportion of patients with MDS. Hence, both classifications yield risk stratification to more comprehensive risk stratification schemes, such as the IPSS-R, and possibly, in the future, to the IPSS-M. However, as more MDS entities become genetically defined, tailored therapy will improve patient outcomes and hopefully, minimize treatment-related toxicities.

The WHO 2016 category therapy-related myeloid neoplasm was removed from both WHO 2022 and ICC 2022 and replaced with diagnostic qualifiers to be added to the relevant MDS subtype. This permits the attribution of germline predisposition to development of MDS and cataloguing of MDS that arose following exposure to certain types of therapy (such as cytotoxic therapy or radiation therapy). This will enhance understanding of issues such as selection pressures of cytotoxic therapy on clonal hematopoiesis and risk of developing MDS.  As more genetic information accumulates concerning MDS that arose after cytotoxic or radiation therapy, this will help answer the questions of whether a subset of these patients can be treated analogous to de novo MDS patients and will permit more tailored therapy for these patients.   Similarly, with increasing awareness and screening for germline predisposition syndromes, it will be important to assess response to therapies in patients with MDS with germline predisposition.

Round 2

Reviewer 1 Report

The manuscript is well written and after some minor revision could be suitable for publication.